# Single-Walled Carbon Nanotubes Attenuate Cytotoxic and Oxidative Stress Response of Pb in Human Lung Epithelial (A549) Cells

**DOI:** 10.3390/ijerph17218221

**Published:** 2020-11-06

**Authors:** Maqusood Ahamed, Mohd Javed Akhtar, M. A. Majeed Khan

**Affiliations:** King Abdullah Institute for Nanotechnology, King Saud University, Riyadh 11451, Saudi Arabia; mjakhtar@ksu.edu.sa (M.J.A.); mmkhan@ksu.edu.sa (M.A.M.K.)

**Keywords:** single-walled carbon nanotubes, Pb exposure, joint toxicity, attenuating effects, A549 cells

## Abstract

Combined exposure of single-walled carbon nanotubes (SWCNTs) and trace metal lead (Pb) in ambient air is unavoidable. Most of the previous studies on the toxicity of SWCNTs and Pb have been conducted individually. There is a scarcity of information on the combined toxicity of SWCNTs and Pb in human cells. This work was designed to explore the combined effects of SWCNTs and Pb in human lung epithelial (A549) cells. SWCNTs were prepared through the plasma-enhanced vapor deposition technique. Prepared SWCNTs were characterized by x-ray diffraction, x-ray photoelectron spectroscopy, scanning electron microscopy, and dynamic light scattering. We observed that SWCNTs up to a concentration of 100 µg/mL was safe, while Pb induced dose-dependent (5–100 µg/mL) cytotoxicity in A549 cells. Importantly, cytotoxicity, cell cycle arrest, mitochondrial membrane potential depletion, lipid peroxidation, and induction of caspase-3 and -9 enzymes following Pb exposure (50 µg/mL for 24 h) were efficiently attenuated by the co-exposure of SWCNTs (10 µg/mL for 24 h). Furthermore, generation of Pb-induced pro-oxidants (reactive oxygen species and hydrogen peroxide) and the reduction of antioxidants (antioxidant enzymes and glutathione) were also mitigated by the co-exposure of SWCNTs. Inductively coupled plasma-mass spectrometry results suggest that the adsorption of Pb on the surface of SWCNTs could attenuate the bioavailability and toxicity of Pb in A549 cells. Our data warrant further research on the combined effects of SWCNTs and Pb in animal models.

## 1. Introduction

Carbon nanotubes (CNTs) have been widely studied due to their distinguished optical, electrical, thermal, and mechanical properties [1,2]. Based on side wall configuration, CNTs can be classified into two groups: single-walled carbon nanotubes (SWCNTs) and multi-walled carbon nanotubes (MWCNTs). SWCNTs comprise of a single sheet of graphene with diameters ranging from 1–3 nm, whereas MWCNTs consist of the collection of nested tubes with increasing diameters ranging from 5–30 nm [3,4]. Due to their distinguished properties, SWCNTs and MWCNTs are being investigated for various applications including optoelectronic, catalysis, energy, bioengineering, biosensors, and drug delivery [5,6,7]. The growing rate in the production and application of CNTs in diverse fields has expected to release a large quantity of SWCNTs and MWCNTs into the environment. The environmental and occupational exposure of CNTs have been previously reported [8,9]. An important study observed that anthropogenic CNTs were found in the airways of Parisian children, indicating that humans are routinely exposed to CNTs [10]. Jung and co-workers suggested that the human lung can be exposed to CNTs through vehicle diesel exhaust [11].

The toxicity of SWCNTs has been previously explored at an individual level in various in vitro and in vivo models [1,12]. Toxicity of SWCNTs might be influenced by several factors including morphology and surface behavior [13,14]. In the real environment, SWCNTs can co-exist with other pollutants such as organic molecules and trace metals. SWCNTs showed strong adsorption affinity to trace metals (e.g., lead (Pb) and cadmium (Cd)) [15]. Hence, interaction of SWCNTs with Pb might affect their bioavailability and toxicity in biological systems. However, there is a critical gap in the knowledge on the combined effects of SCWNTs and Pb in human cells. 

Pb is of global concern because of its hazardous health effects to humans and the environment [16]. Batteries, smelting, paints, glass making, and tobacco smoking are the main sources of Pb exposure [17]. The daily exposure of Pb to ambient air is the second main route of exposure following ingestion [18]. A recent report launched on 30 July 2020 by UNICEF and Pure Earth indicated that up to 800 million children globally (one in three children) had blood Pb level ≥5 µg/mL, a level of health concern. There is no acceptable blood Pb level for the human body and Pb can induce systemic toxicity [19,20]. Oxidative stress and deactivation of sulfhydryl pools of antioxidants are possible mechanisms of Pb toxicity [21].

Co-exposure of SWCNTs and Pb has been reported, and investigations on their joint effects in human cells have been recommended [22,23]. Hence, the purpose of this work was to explore the cytotoxicity and oxidative stress response of pure SWCNTs and Pb individually, and co-exposure in human lung epithelial (A549) cells. In addition, interaction of Pb with SWCNTs in a culture medium was also investigated. We believe that this is the first study to exhibit that Pb-induced cytotoxicity and oxidative stress was effectively attenuated by SWCNTs in A549 cells.

## 2. Materials and Methods

### 2.1. Synthesis of Single-Walled Carbon Nanotubes (SWCNTs)

SWCNTs were prepared by the plasma-enhanced vapor deposition (PECVD) technique as described in our recent study [7]. Brief procedures of SWCNTs are described in Text S1.1 subsection in the Appendix A. 

### 2.2. Characterization of SWCNTs

Crystallinity and phase purity of SWCNTs were assessed by powder x-ray diffraction (PXRD) (PanAnalytic X’Pert Pro, Malvern, UK) with Cu-Kα radiation (λ = 0.15405 nm, at 45 kV and 40 mA). X-ray photoelectron spectroscopy (XPS) analysis was done using an Electron Spectroscopy for Chemical Analysis (ESCA) system (model VG 3000) with monochromatic Mg-Kα line (1253.6 eV) radiation. Morphology was characterized by field emission scanning electron microscopy (FESEM) (JSM-7600F, JEOL, Inc., Tokyo, Japan). Hydrodynamic size and zeta potential of SWCNTs in distilled water and culture medium (DMEM + 10%FBS) were measured by dynamic light scattering (DLS) (Nano-Zeta sizer, Malvern Instruments, Malvern, UK). In brief, SWCNTs (10 µg/mL) were suspended in distilled water and culture media, and sonicated at room temperature for 30 min at 40 W. Then, the DLS measurement was performed at room temperature. We chose a 10 µg/mL concentration of SWCNTs because this concentration was applied in co-exposure experiments.

### 2.3. Cell Culture and Exposure Protocol

Human lung epithelial (A549) cells were cultured in Dulbecco’s modified Eagle’s medium (DMEM) (Invitrogen, Carisbad, CA, USA) with 10% fetal bovine serum (Invitrogen), 100 µg/mL streptomycin, and 100 U/mL penicillin (Invitrogen) at 37 °C with the supply of 5% CO_2_ in a humidified incubator. At 75–85% confluence, cells were harvested with trypsin (Invitrogen) and further cultured for required experiments.

Lead nitrate (Pb(NO_3_)_2_, 99.999%, Sigma-Aldrich, St. Louis, MO, USA) was dissolved in distilled water and used as a source of Pb ions. SWCNTs were also suspended in distilled water. The A549 cells were treated with different concentrations of SWCNTs (0, 1, 5, 10, 25, 50, 100, and 200 µg/mL) and Pb (0, 1, 5, 10, 25, 50, and 100 µg/mL) for 24 h. For a combined toxicity study, we further exposed the cells with either SWCNTs (10 µg/mL), or Pb (50 µg/mL), or a combination of both (10 µg/mL of SWCNTs and 50 µg/mL of Pb) for 24 h. The 10 µg/mL concentration of SWCNTs was chosen on the basis of preliminary screening tests (Appendix A and Text S2 Appendix A).

### 2.4. Biochemical Study

Cell viability was assessed by the MTT (3-(4,5-dimethylthiazol-2-yl)-2,5-diphenyltetrazolium bromide) assay [24] with some essential changes [25]. Cell cycle analysis was done on a Beckman Coulter flow cytometer (Coulter Epics XL/Xl-MCL) through a FL4 filter (585 nm) utilizing propidium iodide (PI, Sigma-Aldrich) [26]. Caspase-3 and -9 enzyme activity was assayed using kits (BioVision, Milpitas, CA, USA). Mitochondrial membrane potential (MMP) was quantified using rhodamine-123 (Rh-123, Sigma-Aldrich) [27]. Probe 2’-7’-dichlorodihydrofluorescein diacetate (H_2_DCFDA, Sigma-Aldrich) was applied to assess ROS generation according to procedures described in a previous study [28]. ROS level was quantitatively analyzed by a microplate reader (Synergy-HT, BioTek Winooski, VT, USA) and intracellular images were captured at a DMi8 fluorescent microscope (Leica Microsystems, Leica Microsystems, GmbH, Germany). Hydrogen peroxide (H_2_O_2_) level was estimated utilizing a kit from Sigma-Aldrich (MAK164). Malondialdehyde (MDA) level was determined using the methods of Ohkawa et al. [29]. Glutathione (GSH) level was assayed as per the instructions of Ellman’s protocol [30]. Rotruck et al.’s [31] procedure was used to assay the glutathione peroxidase (GPx) enzyme. Superoxide dismutase (SOD) enzyme activity was assayed utilizing a kit (Cayman chemical, Ann Arbor, Michigan, USA). Activity of the catalase enzyme was estimated following the method of Sinha [32]. Protein level was assayed using Bradford’s method [33]. Protocols of each experiment were briefly described Text S1 section in the Appendix A.

### 2.5. Inductively Coupled Plasma-Mass Spectrometry

Inductively coupled plasma-mass spectrometry (ICP-MS) was used to assess the adsorption of Pb on the surface of SWCNTs in culture medium as described in our recent paper (Text S1.11 subsection, Appendix A) [34]. Effect of SWCNTs on the cellular uptake of Pb was also measured by ICP-MS (Text S1.12 subsection, Appendix A) [34].

### 2.6. Statistical Analysis

One-way analysis of variance (ANOVA) following Dennett’s multiple comparison tests were applied to analyze the results. The *p* < 0.05 was assigned as statistically significant. 

## 3. Results

### 3.1. Characterization of SWCNTs

Figure 1A shows that the diffraction peaks at 22.78° and 44.36° corresponded to the (002) and (100) planes of the graphitic phase of SWCNTs (JCPDS-75-1621). No peaks were related to impurities or the secondary phase detected in XRD spectra, which indicates the successful synthesis of SWCNTs. The C1s XPS spectra of SWCNTs (Figure 1B) were in agreement with other studies [7]. The morphology of SWCNTs was characterized by FESEM as represented in Figure 1C. This image indicated that uniform SWCNTs was prepared with random orientation. The diameter of SWCNTs was around 1–3 nm, whereas the lengths were in several micrometers.

DLS data are presented in Table 1. The hydrodynamic size of SWCNTs in water and culture media were around 150 nm and 190 nm, respectively. Furthermore, the zeta potential of SWCNTs in water and culture media were approximately −29 mV and −26 mV, respectively. These results suggest that the SWCNTs were fairly stable in distilled water and culture media.

### 3.2. Cell Viability of A549 Cells after Exposure to SWCNTs and Pb Exposure

Cells were treated for 24 h to different concentrations of SWCNTs (0–200 µg/mL) and Pb (0–100 µg/mL). Cell viability was assessed through the MTT assay. Figure 2A shows that SWCNTs did not reduce cell viability up to the concentration of 100 µg/mL. However, at the concentration of 200 µg/mL, SWCNTs caused mild cytotoxicity to A549 cells. On the other hand, Pb-induced cell viability was reduced in a dose-dependent manner (Figure 2B).

### 3.3. SWCNTs Attenuate Pb-Induced Cytotoxicity

Combined cytotoxicity of SWCNTs and Pb was assessed in A549 cells after treatment for 24 to either SWCNTs (10 µg/ml) or Pb (50 µg/mL), or a combination of both (SWCNTs + Pb). Figure 3A exhibited that SWCNTs did not cause cytotoxicity (MTT assay), however, Pb significantly induced cytotoxicity in A549 cells (*p* < 0.05). Interestingly, compared to the Pb group, in the co-exposure group (SWCNTs + Pb), cytotoxicity was reverted and reached almost the level of the SWCNT group or the control group (*p* < 0.05). Flow cytometer results indicated that cell cycle phases in the SWCNT group were similar to the control group (Figure 3B). However, Pb exposure disturbed the cell cycle of A549 cells. Gathering of cells in the sub-G1 phase of the Pb group was significantly higher than those of the control group. Interestingly, in the co-exposure group (SWCNTs + Pb), SWCNTs significantly attenuated the effect of Pb and the number of accumulated cells in the sub-G1 phase reached the level of the SWCNT group or control group. Furthermore, SWCNTs did not affect the activity of the caspase-3 and -9 enzymes, however, Pb exposure significantly increased the activity of these enzymes (*p* < 0.05) (Figure 3C). Noticeably, upon co-exposure (SWCNTs + Pb), SWCNTs effectively alleviated the effects of Pb on the activity of the caspase-3 and -9 enzymes (*p* < 0.05) (Figure 3C). Moreover, Figure 3D shows that SWCNTs did not affect the MMP level, however, Pb significantly decreased the MMP level of A549 cells. Interestingly, in comparison to the Pb group, in the co-exposure group (SWCNTs + Pb), the MMP level increased to the level of the SWCNT group or the control group (*p* < 0.05). Collectively, these data indicated that cytotoxicity exerted by Pb was successfully attenuated by SWCNT co-exposure.

### 3.4. SWCTNs Attenuate Pb-Induced Oxidative Stress

As we can see in Figure 4A, SWCNTs did not induce ROS generation, whereas Pb significantly induced intracellular ROS level in A549 cells (*p* < 0.05). Noticeably, compared to the Pb group, in the co-exposure group (SWCNTs + Pb), the ROS level was significantly decreased to the level of the SWCNT group or control group (*p* < 0.05). Similarly, fluorescent microscopic images of the DCF probe also showed that SWCNTs effectively attenuated Pb-induced intracellular ROS generation in A549 cells (Figure 4B). Furthermore, H_2_O_2_ and MDA (an indicator of membrane lipid peroxidation) levels in the SWCNT group were similar to the control group (Figure 4C,D). However, H_2_O_2_ and MDA levels in the Pb group were significantly higher than those of the control group (*p* < 0.05). Again, in the co-exposure group (SWCNTs + Pb), SWCNTs significantly mitigated the Pb-induced H_2_O_2_ and MDA levels (*p* < 0.05) (Figure 4C,D).

Combined effects of SWCNTs and Pb were further examined on antioxidants markers (Figure 5A–D). The GSH levels and activity of antioxidant enzymes (e.g., GPx, SOD, and CAT) in the SWCNT group were not different from the control group. However, Pb treatment significantly decreased the levels of antioxidants (*p* < 0.05). Noticeably, in comparison to the Pb group, in the co-exposure group (SWCNTs + Pb), GSH, GPx, SOD, and CAT levels increased to a level that was close to the SWCNT group or control group (*p* < 0.05) (Figure 5A–D). Collectively, these results suggest that SWCNTs successfully attenuate Pb-induced oxidative stress in A549 cells.

### 3.5. Inductively Coupled Plasma-Mass Spectrometry Study

Adsorption of Pb on the surface SWCNTs was estimated by ICP-MS. Figure 6A shows that the quantity of free Pb ions in DMEM in the Pb group alone was not much different between 0 h and 24 h. However, in the co-exposure group (SWCNTs + Pb), the level of free Pb ions in DMEM after 24 h was significantly reduced in comparison to 0 h (48.26 µg/mL for 0 h vs. 1.38 µg/mL for 24 h). These results suggest that most of the free Pb ions present in DMEM of the co-exposure group (SWCNTs + Pb) was adsorbed on the surface of SWCNTs.

The effect of SWCNTs on cellular uptake of Pb ions was also determined by ICP-MS. Cells were treated for 24 h to 50 µg/mL of Pb in the presence or absence of SWCNTs (10 µg/mL). Figure 6B shows that the intracellular level of Pb ions in the co-exposure group (SWCNTs + Pb) was significantly lower compared to the Pb group alone. These results suggest that SWCNTs significantly restrict the entry of Pb in A549 cells.

## 4. Discussion

Our environment is a complex system where the exposure of a mixture of unwanted materials such as SWCNTs and Pb is inevitable. Most of the previous studies on the biological interaction of SWCNTs were conducted individually. On the other hand, lead poisoning is a global concern. Pb is a potent neurotoxin that causes irreversible damage to the brains of children [35]. There is a scarcity of information on whether Pb poisoning is influenced by SWCNTs when they are co-exposed in humans. Therefore, this study was conducted to see the whether SWCNTs influence the toxicity of Pb in A549 cells.

The outcomes of this study demonstrated that pure SWCNTs did not reduce cell viability up to the concentration of 100 µg/mL in A549 cells. However, at 200 µg/mL, SWCNTs caused mild cytotoxicity. The toxicity of SWCNTs has been explored for several years, however, conflicting results are available in the literature [36]. Some studies have shown that SWCNTs exert considerable cytotoxicity such as inhibition of cell proliferation, inflammation, oxidative stress, and ultimately, cell death [37,38,39]. However, other reports have suggested that environmentally relevant doses of SWCNTs are safe [40,41]. The toxicity of SWCNTs were credited to size, dose, surface functionalization, and metal impurities [42,43,44]. In this work, prepared SWCNTs were pretreated with calcination to remove any amorphous carbon, and an acid wash (diluted nitric acid) to remove any residual metals. Hence, the effects of these factors on the toxicity of present SWCNTs is likely to be negligible. Our further results showed that SWCNTs did not affect the markers of apoptosis and oxidative stress in A549 cells. 

The results of this work showed that Pb alone induced cell viability reduction, apoptosis, ROS generation, lipid peroxidation, and antioxidant depletion in A549 cells. These results are in agreement with earlier studies [20,45,46]. Interesting findings of this study were that upon co-exposure (SWCNTs + Pb), SWCNTs significantly attenuated the toxicity exerted by Pb exposure in A549 cells.

Cell cycle is a tightly regulated phenomenon that occurs during the process of cell proliferation. Cell cycle consists of four stages: cell enlargement (G1), DNA preparation (S), preparation for cell division (G2), and cell division (M) [47]. Furthermore, genetically damaged cells enter in an apoptotic pathway and gather in the subG1 phase. Our flow-cytometer data with PI probe indicated that Pb-induced disturbance in cell cycle progression was effectively mitigated by SWCNT co-exposure. Pb-induced activity of the caspase-3 and -9 enzymes was also alleviated by SWCNT co-exposure. These caspases belong to a family of proteases that are critically involved in the apoptotic pathway [48]. MMP level decreases under stress response and is an early sign of apoptosis [49]. Our results also demonstrated that a decreased level of MMP due to Pb exposure was significantly reverted by SWCNT co-exposure.

Oxidative stress is the condition where excessive generation of pro-oxidants (e.g., ROS and H_2_O_2_) compromises the antioxidant defense capacity of the cell [50]. ROS have a very short life span and act as signaling agents in the process of apoptosis [51]. Recent research has also suggested that antioxidant GSH depletion also plays a critical role in apoptosis [52,53]. In the present work, we observed that Pb-induced ROS and H_2_O_2_ generation in A549 cells was effectively attenuated by SWCNT co-exposure. Furthermore, depletion of GSH level and lower activity of several antioxidant enzymes (e.g., GPx, SOD, and CAT) after Pb exposure was efficiently abrogated by SWCNT co-exposure. In agreement with the present work, several previous studies have observed that co-exposure of CNTs or nanoparticles might mitigate the toxicity of environmental contaminants. For example, MWCNT co-exposure decreased the toxicity of polycyclic aromatic hydrocarbons (PAHs) in microalga (*Pseudokirchneriella subcapitata*) [54] and soil microbial communities (*alfalfa rhizosphere*) [55]. Another study found that MWCNTs reduced cadmium (Cd) accumulation and toxicity in *Daphnia Magna* [56]. A recent study showed that co-exposure of pure MWCNTs and benzo[a]pyrene (BaP) reduced the cytotoxicity and oxidative stress in comparison to individual exposure in A549 cells [57]. Moreover, our previous investigation also showed that TiO_2_ nanoparticles successfully attenuated Pb-induced cytotoxicity and oxidative stress in A549 cells. Contrary to our data, some studies have reported that the presence of MWCNTs might aggravate the toxicity of trace metals (e.g., Pb and Cd) and organic chemicals (BaP) [58,59]. This could be due to the presence of higher metal impurities in MWCNTS, and probable synergistic toxic interaction between environmental contaminants and metal impurities.

We conducted an ICP-MS study to examine the adoption potential of Pb on the surface of SWCNTs. Results showed that most of the Pb ions present in the culture media were adsorbed on the surface of the SWCNTs. Henceforth, SWCNTs decreased the bioavailability of Pb ions for uptake of A549 cells. Our further results on cellular uptake study demonstrated that due to strong adsorption of Pb ions on the SWCNT surface, the intracellular level of Pb ions in co-exposure group (SWCNTs + Pb) was very low compared to the Pb group alone. This could be one of the potential mechanisms for attenuating the effects of SWCNTs against Pb-induced toxicity. In agreement with our results, Jang and Hwang found that co-exposure of Pb and MWCNTs resulted in decreased Pb toxicity in *Daphnia magna* [60]. The authors suggested that decreased Pb toxicity was due to low bioavailability of free Pb ions caused by a higher adsorption of Pb on the surface of MWCNTs. Our recent studies also showed that reduced graphene oxide (rGO) successfully decreased the toxicity of Pb and Cd in human cells by reducing the bioavailability of metal ions through their adsorption on the rGO surface [61,62]. Underlying mechanisms of the attenuating effect of SWCNTs after trace metal (Pb or Cd) exposure still needs further research.

Understanding the fate and retention time of SWCNTs in cells and living organisms are important points for biomedical application of SWCNTs [63]. Welsher et al. [64] reported that SWCNTs were distributed throughout the body after intravenous injection to mice. Importantly, they observed that no toxic side effects were detected by necropsy, histology, and blood chemistry measurements in those mice in three months after SWCNT injection. However, the fate, retention, and degradation of SWCNTs in cells/organisms in the long-term still remain a daunting task. 

## 5. Conclusions

In conclusion, a benign concentration of SWCNTs (10 µg/mL) successfully attenuated the cytotoxic and oxidative stress response generated by Pb exposure in A549 cells. ICP-MS study demonstrated that due to higher adsorption of Pb on the surface of SWCNTs, cellular uptake of Pb was restricted by SWCNTs. This could be one of the possible mechanisms of attenuating the effects of SWCNTs against Pb-induced toxicity in biological systems. This work warrants further research on the combined effects of SWCNTs and Pb in animal models.

## Figures and Tables

**Figure 1 ijerph-17-08221-f001:**
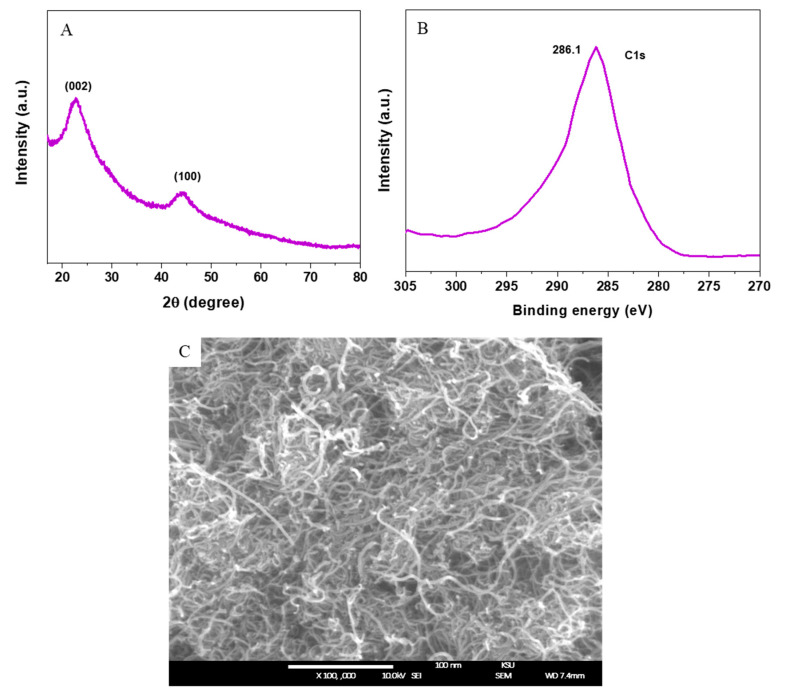
Characterization of single-walled carbon nanotubes (SWCNTs). (**A**) XRD spectra, (**B**) C1s of XPS, and (**C**) FESEM image. SWCNTs: Single-walled carbon nanotubes, XRD: X-ray diffraction, XPS: X-ray photoelectron spectroscopy, FESEM: Field emission electron microscopy.

**Figure 2 ijerph-17-08221-f002:**
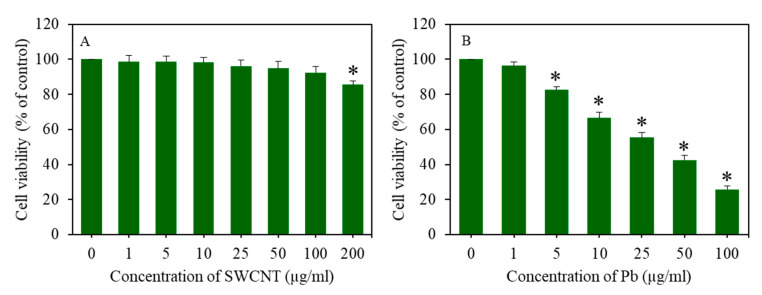
Cell viability of A549 cells after exposure to various concentrations of SWCNTs and Pb for 24 h. (**A**) Cell viability after SWCNT exposure and (**B**) cell viability after Pb exposure. Data provided as mean ± SD of three identical experiments made in three replicates. * Significantly different from the control group (*p* < 0.05). SWCNTs: Single-walled carbon nanotubes, Pb: Lead, A549 cells: Human lung cells.

**Figure 3 ijerph-17-08221-f003:**
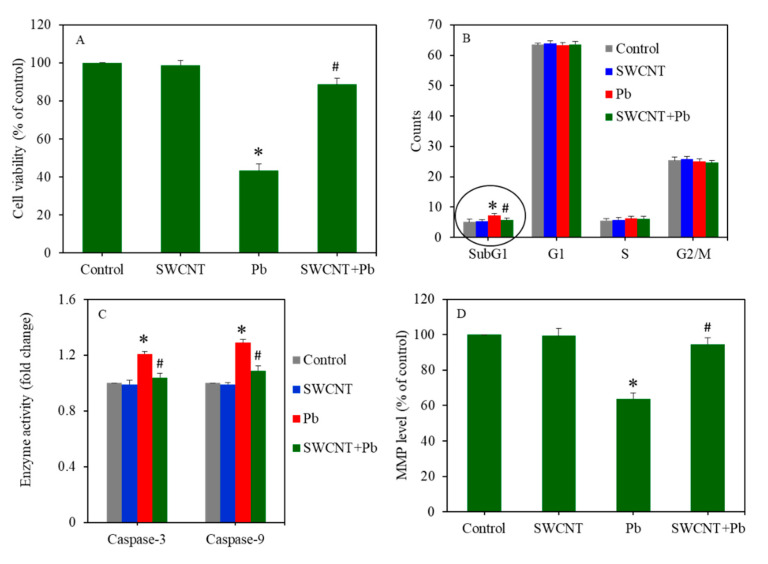
Cytotoxicity endpoints in A549 cells after exposure to either SWCNTs (10 µg/mL) or Pb (50 µg/mL), or a combination of both (SWCNTs + Pb) for 24 h. (**A**) Cell viability, (**B**) cell cycle, (**C**) caspase-3 and -9 enzymes activity, and (**D**) MMP level. Data provided as the mean ± SD of three identical experiments made in three replicates. * Significantly different from the control group (*p* < 0.05). # Attenuating effects of SWCNTs against Pb-induced cytotoxicity. SWCNTs: Single-walled carbon nanotubes, Pb: Lead, A549 cells: Human lung cells, MMP: Mitochondrial membrane potential.

**Figure 4 ijerph-17-08221-f004:**
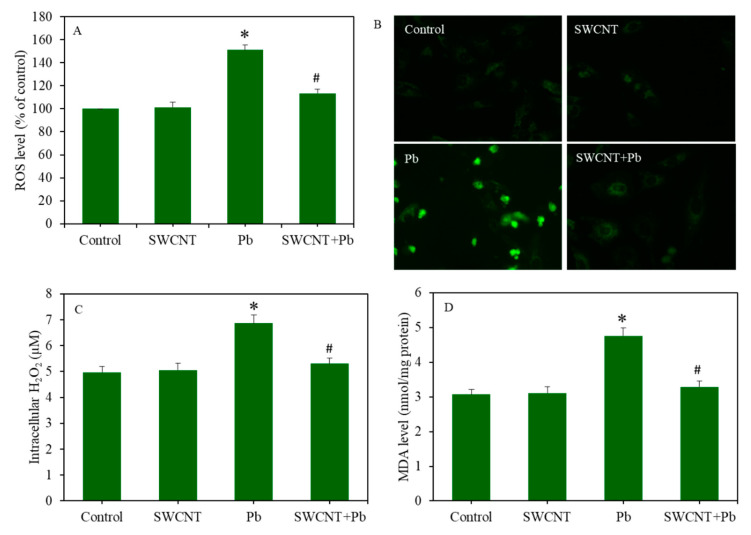
Pro-oxidant levels in A549 cells after exposure to either SWCNTs (10 µg/mL) or Pb (50 µg/mL), or a combination of both (SWCNTs + Pb) for 24 h. (**A**) Quantitative data of intracellular ROS level. (**B**) Fluorescent microscopic images of intracellular ROS level. (**C**) Intracellular H_2_O_2_ level. (**D**) MDA level. Data provided as the mean ± SD of three identical experiments made in three replicates. * Significantly from the control the control group (*p* < 0.05). # Attenuating effects of SWCNTs against Pb-induced pro-oxidant generation. SWCNTs: Single-walled carbon nanotubes, Pb: Lead, A549 cells: Human lung cells, ROS: Reactive oxygen species, H_2_O_2_: Hydrogen peroxide, MDA: Malondialdehyde.

**Figure 5 ijerph-17-08221-f005:**
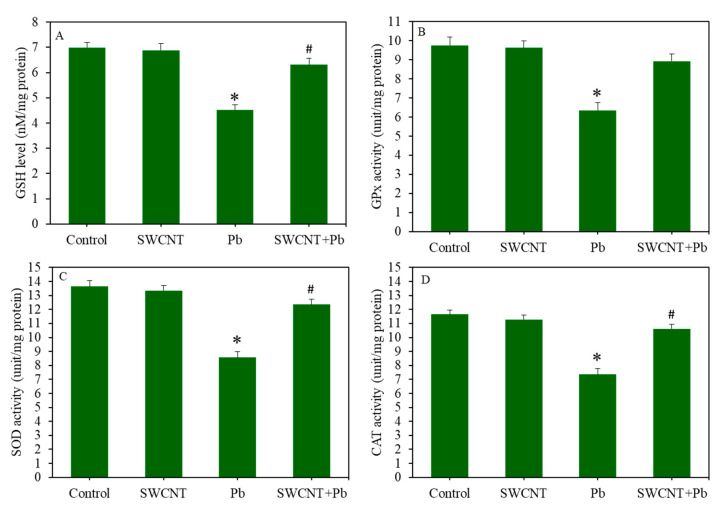
Antioxidant levels in A549 cells after exposure to either SWCNTs (10 µg/mL) or Pb (50 µg/mL), or co-exposure (SWCNTs + Pb) for 24 h. (**A**) Intracellular GSH level. (**B**) GPx enzyme activity. (**C**) SOD enzyme activity. (**D**) CAT enzyme activity. Quantitative data provided in this study are represented are the mean ± SD of three identical experiments made in three replicates. * Significantly different in comparison to the control (*p* < 0.05). # Attenuating effects of SWCNT against Pb-induced antioxidant depletion. SWCNTs: Single-walled carbon nanotubes, Pb: Lead, A549 cells: Human lung cells. GSH: Glutathione, GPx: Glutathione peroxidase, SOD: Superoxide dismutase, CAT: Catalase.

**Figure 6 ijerph-17-08221-f006:**
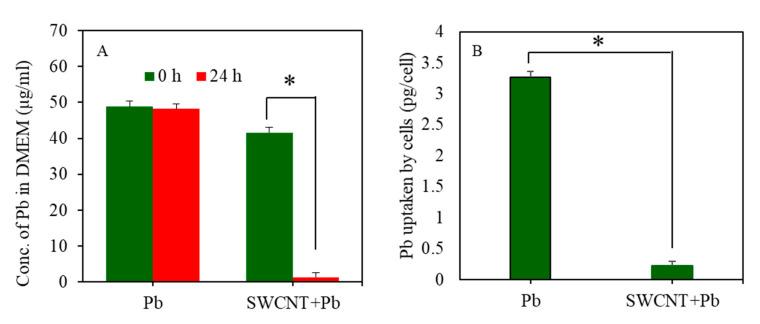
ICP-MS analysis. (**A**) Significant amount Pb present in co-exposure group (SWCNTs + Pb) was adsorbed on the surface of SWCNTs (* *p* < 0.05). (**B**) Significant difference in uptake of Pb by A549 cells between Pb group and co-exposure group (SWCNTs + Pb) (* *p* < 0.05). Pb uptake by cells is presented as a pictogram (pg) of Pb per cell. Data provided as mean ± SD of three identical experiments made in three replicates. SWCNTs: Single-walled carbon nanotubes, Pb: Lead, A549 cells: Human lung cells. ICP-MS: Inductively coupled plasma-mass spectrometry.

**Table 1 ijerph-17-08221-t001:** Dynamic light scattering (DLS) characterization of SWCNTs (mean ± SD, *n* = 3).

Parameters	Hydrodynamic Size nm	Zeta Potential (mV)
Distilled Water	150 ± 13	−29 ± 4
Cultured Medium	170 ± 17	−27 ± 5

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
