# Peer review of "Single-Walled Carbon Nanotubes Attenuate Cytotoxic and Oxidative Stress Response of Pb in Human Lung Epithelial (A549) Cells"

_ijerph, 2020, doi:10.3390/ijerph17218221_

Round 1

Reviewer 1 Report

The revised version is fine for me. I only found minor corrections:

In the main text, line 298 'is still remain' should be modified for 'still remains' without is.

In the supplement, in the title S1.1, the S is missing in SWCNTs.

Reviewer 2 Report

My serious concerns regarding the loopholes in the Methodology are not addressed. Therefore, I am not recommending the manuscript for publication.

Reviewer 3 Report

In the revised manuscript, Ahamed and colleagues effectively addressed the comments raised during the original submission. Therefore, this manuscript can be considered for acceptance in its current form. 

Reviewer 4 Report

I found some minor errors in your manuscript.

Please revise that correctly.  

Fig.2 adjust y axis tick marks and labels (left graph)

Fig.4 H2O2 → H2O2 (left bottom)

Fig.5 proyein → protein  (left upper)

Fig.6 pg/cell → need more explain about experimental method

Author Response

This manuscript is a resubmission of an earlier submission. The following is a list of the peer review reports and author responses from that submission.

Round 1

Reviewer 1 Report

This manuscript described the toxicity of SWCNT and Pb salt alone or combined in a lung cellular model that is pertinent for these pollutants. The manuscript is well written, clear and scientifically sound. The question of co-exposure is very interesting and has to be assessed more and more in the future. In this case, concerning SWCNT and Pb, the results are very interesting since they showed the attenuation of Pb toxicity in presence of SWCNT. The main reason investigated is the adsorption of Pb to SWCNT and this hypothesis is supported by ICP-MS data.

Overall, the manuscript is of good quality and I support its publication to IJERPH.

Only minor corrections should be adjusted:

  • Page 2, line 60-61 : the sentence line 60-61 is not correct grammaticaly
  • Page 9, line 246 : ‘is also plays a critical role’ should be changed by ‘also plays a critical role’.

Reviewer 2 Report

The article titled “Single-Walled Carbon Nanotubes Attenuate Cytotoxic and Oxidative Stress Response of Pb in Human Lung Epithelial (A549) Cells” is within the scope of the International Journal of Environmental Research and Public Health. However, the methodology of the study has serious loopholes.

It is indeed true that in the real environment, CNTs coexist with other toxic compounds because CNTs exhibit a very strong adsorption affinity for various environmental contaminants such as toxic metals. In this case, as the toxic metal, Pb will be adsorbed on the CNTs, and this is well-documented in the literature. As a result, reporting the toxicity of CNTs alone or in combination with Pb, without reporting the adsorption kinetics leaves the study incomplete. Furthermore, since the Pb is expected to be adsorbed on CNTs, a thorough physicochemical characterization will be required. To reach a meaningful conclusion as the authors aimed and stated on lines 58-60: "In order to link the bioavailability and toxicity of Pb with the selected cancer cell line in the presence of SWCNTs", the adsorbed and free Pb concentrations in cell culture conditions (i.e., in the presence and absence of serum proteins) should be reported along with the detailed adsorption kinetics analysis. Again, without this piece of information, solely reporting the toxicity of CNTs alone or in combination with Pb is not meaningful.

The Materials and Methods section needs to be thoroughly revised; all the details regarding the experimental section, including the synthesis of CNTs, need to be explained in great detail. Simply referring to a previous reference is not acceptable (lines 66-67; lines 87-88; lines 99-100; lines 101-102; lines 105-106; lines 117-118).

Characterization with XRD and SEM measurements alone is not adequate to provide a thorough physicochemical characterization analysis. Size/zeta potential and X-ray photoelectron spectroscopy analysis also need to be carried out to link the toxicity results to the physicochemical nature of the material(s) tested.

The concentration range tested in this study for the CNTs needs to be justified (lines 84-85).

In the Results section, for each and every toxicity test that was carried out, all the raw data should be presented in the Supplementary Information. How many statistical replicates were used?

Figure 4B, please provide a higher-resolution image. How were the cells prepared for fluorescence microscopy imaging?

Finally, the language of the main text needs to be significantly improved in terms of grammar, punctuation, and vocabulary.

Reviewer 3 Report

Comment:

In this manuscript, Ahamed and colleagues developed single-walled carbon nanotubes for absorption of Pb ions in biological environment. While Pb ions triggered cytotoxicity, SWCNT did not show significant toxicity when used alone. Meanwhile, SWCNT effectively blocked the cytotoxicity of Pb ions. Various readouts such as cell viability, mitotic activity, intracellular ROS, were characterized to confirm the protective effect of SWCNT. While this is an interesting work, there are still some critical questions that need to be thoroughly addressed (see comments below). These issues necessitate a minor revision to this manuscript before it can be considered for acceptance.

Additional Comments:

  1. Figure 1B needs more information. The nanotube diameter in the image (~10 nm by observation, with some thicker nanotubes ~20 nm) does not correlate with the description (0.8-1.5nm). Please perform quantitative (if possible) analysis on the average diameter of these carbon nanotubes.
  2. The removal efficiency of Pb by SWCNT needs more characterization. For example, in Pb-treated A549 cells, an increasing concentration of SWCNT is added (similar range as Figure 2A to ensure minimal cytotoxicity). Various readouts such as viability could demonstrate the robustness of SWCNT treatment.
  3. The intracellular fate of SWCNT should be studied or discussed. Would these nanotubes stay in the intracellular compartment for prolonged period? If SWCNT can be degraded by cells, would the absorbed Pb ion be released and harm the cells. Please investigate or discuss more.
  4. Please perform a thorough proofread of the manuscript to remove spelling errors.

Reviewer 4 Report

The manuscript by Ahamed et al described that Single-Walled Carbon Nanotubes Attenuate Cytotoxic and Oxidative Stress Response of Pb in Human Lung Epithelial (A549) Cells, proposing that higher adsorption of Pb on the surface of 22 SWCNTs could attenuate the bioavailability and toxicity of Pb in A549 cells.However, Anti-ROS effects  and cell toxicity of SWCNT are known in previous reports, limiting the novelty of this manuscript. In addition, there is no insight into advanced mechanism.